# Synthesis of Non-Aromatic Pyrroles Based on the Reaction of Carbonyl Derivatives of Acetylene with 3,3-Diaminoacrylonitriles

**DOI:** 10.3390/molecules28083576

**Published:** 2023-04-19

**Authors:** Pavel S. Silaichev, Lidia N. Dianova, Tetyana V. Beryozkina, Vera S. Berseneva, Andrey N. Maslivets, Vasiliy A. Bakulev

**Affiliations:** 1Technology of Organic Synthesis Department, Ural Federal University Named after the First President of Russia B. N. Yeltsin, 19 Mira Street, Yekaterinburg 620002, Russia; 2Department of Chemistry, Perm State University, 15 Bukireva Street, Perm 614990, Russia

**Keywords:** 3,3-diaminoacrylonitriles, acetylenedicarboxylate (DMAD), 1,2-dibenzoylacetylene, 5-aminopyrroles, pyrrol-3(2*H*)-ylidene acetates

## Abstract

The reaction of 3,3-diaminoacrylonitriles with DMAD and 1,2-dibenzoylacetylene was studied. It is shown that the direction of the reaction depends on the structure both of acetylene and of diaminoacrylonitrile. In the reaction of DMAD with acrylonitriles bearing a monosubstituted amidine group, 1-substituted 5-amino-2-oxo-pyrrole-3(2*H*)ylidenes are formed. On the other hand, a similar reaction of acrylonitriles containing the *N*,*N*-dialkylamidine group affords 1-*NH*-5-aminopyrroles. In both cases, pyrroles containing two exocyclic double bonds are formed in high yields. A radically different type of pyrroles containing one exocyclic C=C bond and sp^3^ hybrid carbon in the cycle is formed in reactions of 3,3-diaminoacrylonitriles with 1,2-diaroylacetylenes. As in reactions with DMAD, the interaction of 3,3-diaminoacrylonitriles with 1,2-dibenzoylacetylene can lead, depending on the structure of the amidine fragment, both to *NH*- and 1-substituted pyrroles. The formation of the obtained pyrrole derivatives is explained by the proposed mechanisms of the studied reactions.

## 1. Introduction

Esters of acetylenedicarboxylic acid, mostly dimethyl acetylenedicarboxylate (DMAD) (**1**), bearing highly electrophylic triple bond show a broad range of reactivity and are widely used in organic synthesis [1,2,3]. They serve as a two-electron component in formal [2+2]-cycloaddition to arylphosphine oxide to generate four-membered oxaphoshetene [1,2,3]. They can also react with azirines and aziridines as sources of azomethyne ylides to form 4-acylpyrroles [2]. 1,3-Dipolar cycloaddition of DMAD (**1**) to 1,3-dipoles such as azides [2], diazoalkanes [1], nitrile oxides [4], and azomethine ylides [5] affords 1,2,3-triazoles [2], pyrazoles [2], isoxazoles [4,5] and pyrroles [6,7,8]. They can be used in Diels–Alder reactions [9,10] and in reaction with thiolates [11], giving rise to 1:1 adducts and undergoing [8+2]-cycloaddition to form furanophane derivatives [12]. Therefore, DMAD (**1**) can also be involved in a three-component reaction with isocyanides and nucleophilic reagents [13,14,15] to furnish (a) a variety of acyclic and heterocyclic compounds, (b) hybrids of furans with quinolones, and (c) bis(4*H*-chromene)-3,4-dicarboxylate derivatives [15].

In 1998 we discovered that malonothioamides could react with DMAD (**1**) to form 2-oxoethylidene-4-oxothiazolidin-5-ylidenes (Figure 1, path A) [16]. This reaction was also used to prepare various derivatives of 4-oxothiazolidin-5-ylidenes [1,2,3].

We have paid attention to the data of Taran and colleagues [17], who developed a method for the synthesis of 4-arylidene-5-imidazolones based on phosphine-catalyzed tandem umpolung addition and intramolecular cyclization of amidine pronucleophiles with arylpropiolates. Our earlier study of malonothioamides reaction with DMAD (**1**) [16] and the paper of Taran [17] inspired an idea to study the reaction of 3,3-diaminoacrylonitriles **3** with DMAD (**1**) and dibenzoylacetylene (**2**) (Figure 1, Paths D, E).

Cocco and colleagues [18] have reported the formation of ethyl 5-cyano-2-oxo-1,2-dihydropyridine-4-carboxylates in the reaction of 3,3-diaminoacrylonitriles with diethyl 1,2-acetylenedicarboxylate (Figure 1, Path C). Here, we present the formation of *NH*- and 1-substituted pyrrol-3(2)-ylidenes **4** in a similar reaction with DMAD (**1**) (Figure 1, Path D). The present paper also contains the data on our study of the reaction of 3,3-diaminoacrylonitriles **3** with 1,2-dibenzoylacetylene (**2**) leading to novel pyrrole derivatives **5**, bearing an exocyclic C=C bond and an sp^3^-hybride carbon atom (Figure 1, Path E). It is worth noting that in the reactions in Figure 1 (Paths A, B), two heteroatoms (S,N or N,N) of the starting compounds are involved in the structure of products, while in reactions in Paths C, D, E (Figure 1), only one heteroatom of a reagent is incorporated in the product.

## 2. Results and Discussions

### 2.1. Reactions of DMAD (***1***) and 1,2-Dibenzoylacetylene (***2***) with 3,3-Diaminoacrylonitriles ***3***

#### 2.1.1. Synthesis of 2-oxo-1H-pyrrol-3(2*H*)-ylidenes

To study the reactions of DMAD (**1**) and 1,2-dibenzoylacetylene (**2**) with 3,3-diaminoacrylonitriles (2-cyanoacetamidines, **3**), the following starting reagents were used (Figure 1).

It is worth mentioning that Cocco and colleagues have already studied the reaction of 3,3-diaminoacrylonitriles with diethyl 1,2-acetylenedicarboxylates in ethanol and, based on IR and ^1^H NMR spectra, proposed the formation of 2(1*H*)-pyridones [18] (Figure 1, Path C). We have found that DMAD (**1**) smoothly reacts with 3,3-diaminoacrylonitriles **3a m** in DCM at room temperature to form sole products **4a**–**g** or **5a**–**f** at a 71–98% yield (Figure 2).

We have shown the formation of *NH*-pyrroles **4a**–**g** in the reaction of DMAD (**1**) with *N*,*N*-disubstituted 3,3-diaminoacrylonitriles **3a**–**f**. On the other hand, in the reaction of compounds **3g**–**k**,**m** bearing monosubstituted amidine group 1-substituted 5-amino-2-oxopyrrolidenes **5a**–**f** are formed. Interestingly, the reaction of acrylonitrile **3l** with DMAD affords 5-cyclohexylamino-NH-pyrrole **4g** instead of 1-cyclohexylpyrrole **5g** proposed [18]. Probably, the initially formed pyrrole **5g** bearing bulky substituent in position 1 of the ring undergoes the Dimroth rearrangement to form NH-pyrrole **4g**. It should be noted that the data of IR, ^1^H and ^13^C NMR spectra, including 2D NMR spectra obtained for compounds **4b** and **5a,** are in agreement with the structures of 2*H*(1*H*)-pyridones [18]. The final decision in favor of the pyrrole structure came from X-ray data analysis for compound **4e** (Figure 2). These data are in agreement with the formation of a rather 2-oxo-1*H*-pyrrol-3(2*H*)-ylidene acetate structure than of pyridine-2-one ring in this reaction [18]. Thus, we have first demonstrated the formation of 1*H*-pyrrol-3(2*H*)-ylidenes **4** and **5** bearing various primary and secondary amino groups and a variety of substituents (H, alkyl) in position 1 of the ring in the reaction of DMAD (**1**) with 3,3-diaminoacrylonitriles **3a**–**m**.

Figure 3 illustrates the plausible mechanism for the formation of pyrroles **4** and **5** in the reaction of 3,3-diaminoacrylonitriles **3** with DMAD (**1**).

It is reasonable to assume that the initial addition of a highly electrophilic alkyne group of DMAD (**1**) to position 2 of 2-cyanoacetamidine **A** results in the formation of intermediate **B**. The nucleophilicity of A is increased by the *tert*-amino effect of the amino group (Figure 3). Then, the rotation around the single bond in the intermediate occurs. It is followed by an H-shift in **C**, generating key intermediate **D**. Interaction of ester and amino groups finalizes the process of formation of either *NH*-pyrrole **4** (when R^3^ = H) or 1-substituted pyrrole **5** (in the case when R^3^ ≠ H).

Pyrroles are important scaffolds due to their presence in various biologically active naturally occurring compounds (porphyrin, hemoglobin, chlorophyll, Vitamin B12). These compounds exhibit anti-inflammatory, antioxidant, and anticancer activities [19]. With the purpose of expanding the scope of pyrrole derivatives prepared from 3,3-diaminoacrylonitriles **3**, we have also carried out a detailed study of the reactions of 3,3-diaminoacrylonitriles **3a**–**g**, **k**, **l** with 1,2-dibenzoylacetylene (**2**). To the best of our knowledge, the reaction of 3,3-diaminoacrylonitriles **3** with 1,2-dibenzoylacetylene (**2**) was not studied so far.

#### 2.1.2. Synthesis of 5-Hydroxypyrroles

Similar to the reaction of DMAD (**1**), the reaction of 3,3-diaminoacrylonitriles **3** with 1,2-dibenzoylacetylene (**2**) in DCM leads to the formation of two types of products (depending on the structure of amidine group), 1-nonsubstituted *NH*-pyrroles **6a**–**g** or *N*-substituted pyrroles **7a**,**b** in 83–98 and in 76–92% yield, respectively (Figure 4). The structure of the prepared compounds is in good agreement with IR, ^1^H and ^13^C NMR spectra, including the data of HSQC and HMBC NMR spectra of compound **6b** (Appendix A) and with the data of high-resolution mass spectrometry (HRMS). The final proof of the structure of compounds **6** and **7** came from X-ray data analysis for compounds **6e** and **7a** (Figure 4). Thus, we have elaborated on an effective novel method of the synthesis of nonaromatic pyrroles bearing C=C bond and sp^3^ hybrid carbon atom in the ring.

The mechanism of formation of pyrroles **6** and **7** (Figure 5) is similar to that of pyrroles **4** and **5** (Figure 2). The first C–C bond in both compounds **6** and **7** is formed similarly to the formation of compounds **3** and **4** via the interaction of a negatively charged carbon atom of intermediate **A** with a triple bond of dibenzoylacetylene (**2**). The N–C bond of pyrroles **6** and **7** is formed via the addition of an amino group to the C=O bond in intermediate **D**. It is different from the mechanism depictured in Figure 2, where the N–C bond is formed via the interaction of ester and amino groups.

## 3. Materials and Methods

All chemicals were purchased from commercial sources and were used without further purification. Analytical thin-layer chromatography was performed on aluminium foil plates Sorbfil UV-254 coated with 0.2 mm silica gel and visualized with UV-lamp 254 nm in an EtOAc/petroleum ether (PE) system (3:1, 2:1 or 1:2). Melting points were determined on a melting point apparatus Stuart SMP10 (Staffordshire, ST15 OSA, UK) and are uncorrected. All NMR spectra were recorded with a Bruker Avance II (Karlsruhe, Germany) spectrometer at 400 MHz, 600 MHz (^1^H NMR) and 100 MHz (^13^C NMR) in CDCl_3_ and DMSO-*d*_6_. The chemical shifts are given in ppm relative to the resonance of the solvents [^1^H: δ (CHCl_3_) = 7.26, ^13^C: *δ* (CDCl_3_) = 77.16 ppm; ^1^H: *δ* (DMSO-*d*_5_) = 2.50, ^13^C: *δ* (DMSO-*d*_6_) = 39.52 ppm]. Multiplicities were given as: s (singlet); br. s (broad singlet); d (doublet); t (triplet); dd (double of doublet); m (multiplet). Coupling constants are reported as *J* value in Hz. The minor isomer signal is highlighted with an asterisk (*). High-resolution mass spectra (HRMS) were recorded using ultrahigh resolution quadrupole time-of-flight mass spectrometer Bruker maXis impact HD (Billerica, MA, USA) with the electrospray ionization probe coupled with Agilent 1260 HPLC system. The Fourier-transform infrared (FT-IR) spectra were obtained using a Bruker Alpha (ATR, ZnSe) spectrometer (Ettlingen, Germany) in the 4000–500 cm^–1^ region.

### 3.1. Synthesis

#### 3.1.1. Preparation of 3,3-Diaminoacrylonitriles **3**

3,3-Diaminoacrylonitriles **3a**–**f**, **h**, **l** were synthesized from ethyl 2-cyanoacetimidate and corresponding amines according to the literature procedures [20,21,22,23]; the compounds **3g**, **i**–**k**, **m** are commercially available.

#### 3.1.2. Synthesis of Pyrroles **4a**–**g**, **5a**–**f**. General Procedure

DMAD (**1**) (0.5 mmol, 71 mg) was added to the solution of corresponding 3,3-diaminoacrylonitryle **3** (0.5 mmol) in DCM (2 mL) at room temperature. The reaction mixture was stirred for 30 min at room temperature, then PE (10 mL) was added, and the resulting solution was stirred for 5 min more. The formed precipitate was filtered off, washed with DCM/PE (1:5) and dried.

*Methyl (E)-2-(4-cyano-5-(dimethylamino)-2-oxo-1,2-dihydro-3H-pyrrol-3-ylidene)acetate* (**4a**). Compound **4a** was obtained at a 98% yield (109 mg), according to the general procedure (amidine **3a**: 56 mg, 0.5 mmol; acetylene **1**: 71 mg, 0.5 mmol; DCM (2 mL)) as a yellow solid, mp 241–243 °C. ^1^H NMR (400 MHz, DMSO-*d*_6_): *δ* 3.29 (s, 6H), 3.64 (s, 3H), 5.71 (s, 1H), 11.27 (br. s, 1H). ^13^C NMR (100 MHz, DMSO-*d*_6_): *δ* 40.7, 50.4, 60.7, 102.4, 117.4, 139.1, 161.5, 165.8, 168.3. IR (ATR, ZnSe, cm^−1^): *ν*, 3131, 3057, 2776, 2202, 1744, 1696, 1637, 1596, 1449, 1395, 1352, 1297, 1154, 1137, 1011. HRMS (ESI-TOF) *m*/*z*: [M + H]^+^ Calcd. for C_10_H_12_N_3_O_3_ 222.0873; Found: 222.0880.

*Methyl (E)-2-(4-cyano-2-oxo-5-(pyrrolidin-1-yl)-1,2-dihydro-3H-pyrrol-3-ylidene)acetate* (**4b**). Compound **4b** was obtained at a 79% yield (97 mg), according to the general procedure (amidine **3b**: 69 mg, 0.5 mmol; acetylene **1**: 71 mg, 0.5 mmol; DCM (2 mL)) as a yellow solid, mp 254–255 °C. ^1^H NMR (400 MHz, DMSO-*d*_6_): *δ* 1.91–1.98 (m, 4H), 3.57 (br. s, 2H), 3.63 (s, 3H), 3.88 (br. s, 2H), 5.66 (s, 1H), 11.32 (br. s, 1H). ^13^C NMR (100 MHz, DMSO-*d*_6_): *δ* 24.0, 25.2, 49.2, 50.4, 50.9, 60.9, 101.7, 117.5, 139.0, 158.8, 166.0, 168.5. IR (ATR, ZnSe, cm^−1^): *ν*, 3131, 3057, 2776, 2202, 1744, 1696, 1637, 1596, 1449, 1395, 1352, 1297, 1154, 1137, 1011. HRMS (ESI-TOF) *m*/*z*: [M + H]^+^ Calcd. for C_12_H_14_N_3_O_3_ 248.1030; Found: 248.1028.

*Methyl (E)-2-(4-cyano-2-oxo-5-(piperidin-1-yl)-1,2-dihydro-3H-pyrrol-3-ylidene)acetate* (**4c**). Compound **4c** was obtained at a 97% yield (128 mg), according to the general procedure (amidine **3c**: 76 mg, 0.5 mmol; acetylene **1**: 71 mg, 0.5 mmol; DCM (2 mL)) as a yellow solid, mp 216–218 °C. ^1^H NMR (400 MHz, DMSO-*d*_6_): *δ* 1.66 (br. s, 6H), 3.64 (s, 3H), 3.76 (br. s, 4H), 5.71 (s, 1H), 11.34 (br. s, 1H). ^13^C NMR (100 MHz, DMSO-*d*_6_): *δ* 23.1, 25.7, 49.4, 50.5, 60.7, 102.4, 117.2, 139.3, 160.2, 165.8, 168.4. IR (ATR, ZnSe, cm^−1^): *ν*, 3271, 3062, 2929, 2854, 2196, 1744, 1694, 1634, 1589, 1558, 1447, 1368, 1256, 1151, 1020. HRMS (ESI-TOF) *m*/*z*: [M + H]^+^ Calcd. for C_13_H_16_N_3_O_3_ 262.1186; Found: 262.1188.

*Methyl (E)-2-(5-(4-benzylpiperidin-1-yl)-4-cyano-2-oxo-1,2-dihydro-3H-pyrrol-3-ylidene)acetate* (**4d**). Compound **4d** was obtained at an 83% yield (146 mg), according to the general procedure (amidine **3d**: 121 mg, 0.5 mmol; acetylene **1**: 71 mg, 0.5 mmol; DCM (2 mL)) as a yellow solid, mp 232–234 °C. ^1^H NMR (400 MHz, DMSO-*d*_6_): *δ* 1.27–1.37 (m, 2H), 1.70–1.73 (m, 2H), 1.86–1.97 (m, 1H), 2.53 (d, *J* = 8 Hz, 2H), 3.22–3.25 (m, 2H), 3.64 (s, 3H), 4.26 (br. s, 2H), 5.73 (s, 1H), 7.17–7.21 (m, 3H), 7.27–7.31 (m, 2H), 11.29 (br. s, 1H). ^13^C NMR (100 MHz, DMSO-*d*_6_): *δ* 31.5, 36.2, 41.4, 48.4, 50.4, 60.8, 102.7, 117.0, 125.9, 128.1, 128.9, 139.1, 139.6, 160.2, 165.7, 168.3. IR (ATR, ZnSe, cm^−1^): *ν* 3149, 3083, 2918, 2200, 1718, 1688, 1627, 1597, 1493, 1452, 1375, 1281, 1185, 1153, 1111, 1091, 1072, 1040. HRMS (ESI-TOF) *m*/*z*: [M + Na]^+^ Calcd. for C_20_H_21_N_3_NaO_3_ 374.1475; Found: 374.1472.

*Methyl (E)-2-(5-(azepan-1-yl)-4-cyano-2-oxo-1,2-dihydro-3H-pyrrol-3-ylidene)acetate* (**4e**). Compound **4e** was obtained at an 82% yield (113 mg), according to the general procedure (amidine **3e**: 83 mg, 0.5 mmol; acetylene **1**: 71 mg, 0.5 mmol; DCM (2 mL)) as a yellow solid, mp 235–236 °C. ^1^H NMR (400 MHz, DMSO-*d*_6_): *δ* 1.55 (br. s, 4H), 1.64–1.80 (m, 4H), 3.64 (s, 3H), 3.68–3.95 (m, 4H), 5.72 (s, 1H), 11.25 (br. s, 1H). ^13^C NMR (100 MHz, DMSO-*d*_6_): *δ* 25.7, 27.7, 50.4, 51.2, 60.1, 102.6, 117.1, 138.9, 160.4, 165.8, 168.3. IR (ATR, ZnSe, cm^−1^): *ν* 3159, 3089, 2923, 2857, 2184, 1725, 1688, 1631, 1585, 1451, 1403, 1355, 1261, 1158, 1097, 1045. HRMS (ESI-TOF) *m*/*z*: [M + H]^+^ Calcd. for C_14_H_18_N_3_O_3_ 276.1343; Found: 276.1347.

*Methyl (E)-2-(4-cyano-5-morpholino-2-oxo-1,2-dihydro-3H-pyrrol-3-ylidene)acetate* (**4f**). Compound **4f** was obtained at a 76% yield (100 mg), according to the general procedure (amidine **3f**: 77 mg, 0.5 mmol; acetylene **1**: 71 mg, 0.5 mmol; DCM (2 mL)) as a yellow solid, mp 235–237 °C. ^1^H NMR (400 MHz, DMSO-*d*_6_): *δ* 3.65 (s, 3H), 3.73–3.75 (m, 4H), 3.78–3.80 (m, 4H), 5.78 (s, 1H); 11.35 (br. s, 1H). ^13^C NMR (100 MHz, DMSO-*d*_6_): *δ* 48.3, 50.5, 61.0, 65.5, 103.6, 117.0, 138.8, 160.8, 165.7, 168.1. IR (ATR, ZnSe, cm^−1^): *ν* 3184, 2981, 2875, 2187, 1755, 1713, 1700, 1575, 1461, 1432, 1357, 1297, 1266, 1202, 1145, 1114, 1066, 1002. HRMS (ESI-TOF) *m*/*z*: [M + Na]^+^ Calcd. for C_12_H_13_N_3_NaO_4_ 286.0798; Found: 286.0800.

*Methyl (E)-2-(4-cyano-5-(cyclohexylamino)-2-oxo-1,2-dihydro-3H-pyrrol-3-ylidene)acetate* (**4g**). Compound **4g** was obtained at an 89% yield (122 mg), according to the general procedure (amidine **3l**: 83 mg, 0.5 mmol; acetylene **1**: 71 mg, 0.5 mmol; DCM (2 mL)) as a yellow solid, mp 236–237 °C. ^1^H NMR (400 MHz, DMSO-*d*_6_): *δ* 1.04–1.11 (m, 1H), 1.21–1.31 (m, 2H), 1.41–1.49 (m, 2H), 1.57–1.60 (m, 1H), 1.70–1.82 (m, 4H), 3.53 (br. s, 1H), 3.63 (s, 3H), 5.60 (s, 1H), 8.78 (br. s, 1H), 11.45 (br. s, 1H). ^13^C NMR (100 MHz, DMSO-*d*_6_): *δ* 24.4, 24.5, 32.1, 50.2, 53.5, 60.8, 100.1, 116.0, 138.0, 161.9, 166.3, 169.1. IR (ATR, ZnSe, cm^−1^): *ν* 3210, 3139, 3026, 2951, 2937, 2859, 2199, 1731, 1705, 1650, 1598, 1517, 1452, 1439, 1398, 1357, 1334, 1306, 1272, 1258, 1192, 1171, 1150, 1123, 1077, 1053, 1044. HRMS (ESI-TOF) *m*/*z*: [M + Na]^+^ Calcd. for C_14_H_17_N_3_NaO_3_ 298.1162; Found: 298.1159.

*Methyl (E)-2-(5-amino-1-benzyl-4-cyano-2-oxo-1,2-dihydro-3H-pyrrol-3-ylidene)acetate* (**5a**). Compound **5a** was obtained at a 72% yield (102 mg), according to the general procedure (amidine **3g**: 87 mg, 0.5 mmol; acetylene **1**: 71 mg, 0.5 mmol; DCM (2 mL)) as a yellow solid, mp 192–193 °C. ^1^H NMR (400 MHz, DMSO-*d*_6_): *δ* 3.66 (s, 3H), 4.88 (s, 2H), 5.76 (s, 1H), 7.18–7.28 (m, 2H), 7.30–7.37 (m, 3H), 8.97 (br. s, 2H). ^13^C NMR (100 MHz, DMSO-*d*_6_): *δ* 42.1, 50.5, 60.5, 102.7, 115.7, 126.7, 127.5, 128.6, 135.7, 136.6, 163.7, 166.1, 167.6. IR (ATR, ZnSe, cm^−1^): *ν* 3140, 3117, 2991, 2198, 1736, 1707, 1651, 1607, 1555, 1495, 1441, 1408, 1362, 1317, 1169, 1118, 1079, 1047, 1025. HRMS (ESI-TOF) *m*/*z*: [M + Na]^+^ Calcd. for C_15_H_13_N_3_NaO_3_ 306.0849; Found: 306.0847.

*Methyl (E)-2-(5-amino-4-cyano-1-(2,4-difluorobenzyl)-2-oxo-1,2-dihydro-3H-pyrrol-3-ylidene)acetate* (**5b**). Compound **5b** was obtained at a 98% yield (78 mg), according to the general procedure (amidine **3h**: 105 mg, 0.5 mmol; acetylene **1**: 71 mg, 0.5 mmol; DCM (2 mL)) as a yellow solid, mp 208–210 °C. ^1^H NMR (600 MHz, DMSO-*d*_6_): *δ* 3.66/3.61* (s, 3H), 4.89/4.82* (s, 2H), 5.74/5.69* (s, 1H), 7.03–7.07 (m, 1H), 7.13–7.19 (m, 1H), 7.24–7.29 (m, 1H), 8.97 (br. s, 2H). ^13^C NMR (100 MHz, DMSO-*d*_6_): *δ* 37.1/37.0*, 50.5, 60.7, 102.7, 104.1 (t, *J* = 25.7 Hz), 111.5 (dd, *J* = 21.2, 3.5 Hz), 115.7, 118.9 (dd, *J* = 14.7, 3.6 Hz), 129.3 (dd, *J* = 10.0, 5.6 Hz), 136.5, 159.8 (dd, *J* = 248.4, 12.4 Hz), 161.7 (dd, *J* = 246.2, 12.2 Hz), 163.6, 166.1, 167.4. IR (ATR, ZnSe, cm^−1^): *ν* 3153, 2947, 2824, 2197, 1742, 1691, 1632, 1591, 1436, 1421, 1376, 1279, 1163, 1080, 1084, 1072, 1065, 1044. HRMS (ESI-TOF) *m*/*z*: [M + H]^+^ Calcd. for C_15_H_12_F_2_N_3_O_3_ 320.0841; Found: 320.0843.

*Methyl (E)-2-(5-amino-4-cyano-2-oxo-1-propyl-1,2-dihydro-3H-pyrrol-3-ylidene)acetate (***5c**). Compound **5c** was obtained at a 71% yield (83 mg), according to the general procedure (amidine **3i**: 63 mg, 0.5 mmol; acetylene **1**: 71 mg, 0.5 mmol; DCM (2 mL)) as a yellow solid, mp 182–183 °C. ^1^H NMR (400 MHz, DMSO-*d*_6_): *δ* 0.82/0.86* (t, *J* = 7.2 Hz, 3H), 1.46–1.51 (m, 2H), 3.56 (t, *J* = 7.2 Hz, 2H), 3.65/3.63* (s, 3H), 5.72 (s, 1H), 8.82 (br. s, 2H). ^13^C NMR (100 MHz, DMSO-*d*_6_): *δ* 10.6, 21.2/22.7*, 40.5, 50.4/50.3*, 60.1, 102.2, 115.8, 136.8, 163.9, 166.1, 167.5. IR (ATR, ZnSe, cm^−1^): *ν* 3320, 3279, 3236, 3198, 3169, 2969, 2951, 2879, 2197, 1748, 1730, 1706, 1658,1614, 1561, 1499, 1451, 1412, 1385, 1348, 1318, 1277, 1199, 1177, 1115, 1051, 1042, 1025. HRMS (ESI-TOF) *m*/*z*: [M + H]^+^ Calcd. for C_11_H_14_N_3_O_3_ 236.1030; Found: 236.1027.

*Methyl (E)-2-(1-allyl-5-amino-4-cyano-2-oxo-1,2-dihydro-3H-pyrrol-3-ylidene)acetate* (**5d**). Compound **5d** was obtained at a 94% yield (110 mg), according to the general procedure (amidine **3j**: 62 mg, 0.5 mmol; acetylene **1**: 71 mg, 0.5 mmol; DCM (2 mL)) as a yellow solid, mp 176–177 °C. ^1^H NMR (400 MHz, DMSO-*d*_6_): *δ* 3.66/3.64* (s, 3H), 4.25–4.27 (m, 2H), 5.00–5.05 (m, 1H), 5.12–5.15 (m, 1H), 5.73–5.83 (m, 2H), 8.81 (br. s, 2H). ^13^C NMR (100 MHz, DMSO-*d*_6_): *δ* 40.9, 50.4, 60.2, 102.5, 115.7, 116.3, 131.6/133.5*, 136.6, 163.6, 166.1, 167.2. IR (ATR, ZnSe, cm^−1^): *ν* 3395, 3316, 3279, 3238, 3199, 3169, 2950, 2197, 1749, 1730, 1706, 1697, 1657, 1613, 1562, 1494, 1448, 1411, 1385, 1319, 1201, 1183, 1136, 1116, 1050. HRMS (ESI-TOF) *m*/*z*: [M + H]^+^ Calcd. for C_11_H_12_N_3_O_3_ 234.0873; Found: 234.0853.

*Methyl (E)-2-(5-amino-4-cyano-2-oxo-1-(prop-2-yn-1-yl)-1,2-dihydro-3H-pyrrol-3-ylidene)acetate* (**5e**). Compound **5e** was obtained at a 96% yield (111 mg), according to the general procedure (amidine **3k**: 61 mg, 0.5 mmol; acetylene **1**: 71 mg, 0.5 mmol; DCM (2 mL)) as a yellow solid, mp 202–203 °C. ^1^H NMR (400 MHz, DMSO-*d*_6_): *δ* 3.66 (s, 3H), 4.47 (d, *J* = 2.2 Hz, 2H), 5.76 (s, 1H), 8.97 (br. s, 2H). ^13^C NMR (100 MHz, DMSO-*d*_6_): *δ* 28.9, 50.5, 60.7, 74.8, 77.3, 103.0, 115.4, 136.3, 162.7, 165.9, 166.7. IR (ATR, ZnSe, cm^−1^): *ν* 3382, 3301, 3281, 3238, 3205, 3175, 2950, 2201, 2193, 2129, 1753, 1704, 1665, 1618, 1563, 1498, 1450, 1427, 1408, 1385, 1323, 1305, 1197, 1182, 1167, 1137, 1089. HRMS (ESI-TOF) *m*/*z*: [M + H]^+^ Calcd. for C_11_H_10_N_3_O_3_ 232.0717; Found: 232.0710.

*Methyl (E)-2-(5-amino-4-cyano-1-(2,2-dimethoxyethyl)-2-oxo-1,2-dihydro-3H-pyrrol-3-ylidene)acetate* (**5f**). Compound **5f** was obtained at a 92% yield (129 mg), according to the general procedure (amidine **3m**: 86 mg, 0.5 mmol; acetylene **1**: 71 mg, 0.5 mmol; DCM (2 mL)) as a yellow solid, mp 178–179 °C. ^1^H NMR (400 MHz, DMSO-*d*_6_): *δ* 3.29 (s, 6H), 3.65 (s, 3H), 3.76 (d, *J* = 5.5 Hz, 2H), 4.50 (t, *J* = 5.5 Hz, 1H), 5.73 (s, 1H), 8.78 (br. s, 2H). ^13^C NMR (100 MHz, DMSO-*d*_6_): *δ* 40.9, 50.4, 54.3, 60.4, 100.7, 102.4, 115.6, 136.5, 163.9, 166.1, 167.5. IR (ATR, ZnSe, cm^−1^): *ν* 3371, 3293, 3141, 3006, 2962, 2944, 2841, 2801, 2202, 1756, 1706, 1686, 1623, 1567, 1502, 1464, 1435, 1416, 1402, 1385, 1361, 1318, 1219, 1198, 1175, 1134, 1100, 1037, 1006. HRMS (ESI-TOF) *m*/*z*: [M + H]^+^ Calcd. for C_12_H_16_N_3_O_5_ 282.1084; Found: 282.1089.

#### 3.1.3. Synthesis of Pyrroles **6a**–**g**, **7a**,**b**. General Procedure

Corresponding 3,3-diaminoacrylonitryle **3** (0.5 mmol) was added to the solution of dibenzoylacetylene **2** (0.5 mmol, 117 mg) in DCM (2 mL) at room temperature. The reaction mixture was stirred for 30 min at room temperature, then ethanol (4 mL) was added, and the resulting solution was stirred for 5 min more. The formed precipitate was filtered off, washed with cold ethanol (1:5) and dried.

*(Z)-2-(Dimethylamino)-5-hydroxy-4-(2-oxo-2-phenylethylidene)-5-phenyl-4,5-dihydro-1H-pyrrole-3-carbonitrile* (**6a**). Compound **6a** was obtained at a 96% yield (158 mg), according to the general procedure (amidine **3a**: 56 mg, 0.5 mmol; acetylene **2**: 117 mg, 0.5 mmol; DCM (2 mL)) as a yellow solid, mp 162–164 °C. ^1^H NMR (400 MHz, CDCl_3_): *δ* 3.26 (br. s, 6H), 5.78 (br. s, 1H), 6.65 (s, 1H), 7.26–7.36 (m, 5H), 7.41–7.45 (m, 1H), 7.59 (d, *J* 8.0 Hz, 2H), 7.82 (d, *J* 8.0 Hz, 2H), 9.24 (br. s, 1H). ^13^C NMR (100 MHz, CDCl_3_): *δ* 40.2, 68.4, 92.0, 101.4, 117.4, 124.9, 128.3, 128.4, 128.6, 128.8, 132.0, 139.4, 141.4, 161.9, 169.6, 188.9. IR (ATR, ZnSe, cm^−1^): *ν* 3414, 3228, 3059, 3025, 2937, 2190, 1638, 1596, 1573, 1521, 1458, 1432, 1400, 1329, 1307, 1218, 1199, 1177, 1158, 1132, 1069, 1047, 1024, 1001. HRMS (ESI-TOF) *m*/*z*: [M + H]^+^ Calcd. for C_21_H_20_N_3_O_2_ 346.1550; Found: 346.1545.

*(Z)-5-Hydroxy-4-(2-oxo-2-phenylethylidene)-5-phenyl-2-(pyrrolidin-1-yl)-4,5-dihydro-1H-pyrrole-3-carbonitrile* (**6b**). Compound **6b** was obtained at a 97% yield (170 mg), according to the general procedure (amidine **3b**: 70 mg, 0.5 mmol; acetylene **2**: 117 mg, 0.5 mmol; DCM (2 mL)) as a yellow solid, mp 205–206 °C. ^1^H NMR (400 MHz, CDCl_3_): *δ* 1.88–1.94 (m, 4H), 3.18 (br. s, 1H), 3.35 (br. s, 1H), 3.90 (br. s, 2H), 6.44 (s, 1H), 6.51 (s, 1H), 7.19–7.27 (m, 5H), 7.34 (t, 1H, *J* = 7.2 Hz), 7.50 (d, 2H, *J* = 6.7 Hz), 7.71 (d, 2H, *J* = 7.4 Hz), 8.45 (br. s, 1H). ^13^C NMR (100 MHz, CDCl_3_): *δ* 24.8, 25.9, 49.2, 68.9, 92.4, 100.3, 117.4, 125.1, 128.2, 128.3, 128.4, 128.5, 131.8, 139.6, 141.5, 159.1, 170.2, 188.3. IR (ATR, ZnSe, cm^−1^): *ν* 3419, 3268, 3216, 3171, 3057, 2991, 2960, 2881, 2200, 1639, 1594, 1568, 1524, 1490, 1451, 1432, 1386, 1355, 1325, 1301, 1244, 1230, 1214, 1194, 1180, 1156, 1137, 1111, 1084, 1072, 1065, 1048, 1025. HRMS (ESI-TOF) *m*/*z*: [M + H]^+^ Calcd. for C_23_H_22_N_3_O_2_ 372.1707; Found: 372.1704.

*(Z)-5-Hydroxy-4-(2-oxo-2-phenylethylidene)-5-phenyl-2-(piperidin-1-yl)-4,5-dihydro-1H-pyrrole-3-carbonitrile* (**6c**). Compound **6c** was obtained at a 98% yield (180 mg), according to the general procedure (amidine **3c**: 76 mg, 0.5 mmol; acetylene **2**: 117 mg, 0.5 mmol; DCM (2 mL)) as a yellow solid, mp 148–150 °C. ^1^H NMR (400 MHz, CDCl_3_): *δ* 1.63 (s, 6H), 3.46–3.67 (m, 4H), 6.36 (s, 1H), 6.55 (br. s, 1H), 7.19–7.28 (m, 5H), 7.35 (t, *J* = 7.4 Hz, 1H), 7.51 (d, *J* 6.8 Hz, 2H), 7.73 (d, *J* = 7.2 Hz, 2H,), 8.48 (br. s, 1H). ^13^C NMR (100 MHz, CDCl_3_): *δ* 23.7, 25.9, 49.2, 68.6, 91.9, 100.6, 117.3, 125.0, 128.3, 128.3, 128.5, 128.7, 131.9, 139.5, 141.4, 160.5, 170.3, 188.5. IR (ATR, ZnSe, cm^−1^): *ν* 3418, 3195, 3061, 3031, 2941, 2858, 2189, 1622, 1595, 1569, 1519, 1491, 1450, 1435, 1400, 1385, 1355, 1326, 1300, 1232, 1219, 1198, 1178, 1131, 1085, 1071, 1051, 1022, 1004. HRMS (ESI-TOF) *m*/*z*: [M + H]^+^ Calcd. for C_24_H_24_N_3_O_2_ 386.1863; Found: 386.1861.

*(Z)-2-(4-Benzylpiperidin-1-yl)-5-hydroxy-4-(2-oxo-2-phenylethylidene)-5-phenyl-4,5-dihydro-1H-pyrrole-3-carbonitrile* (**6d**). Compound **6d** was obtained at an 83% yield (198 mg), according to the general procedure (amidine **3d**: 120 mg, 0.5 mmol; acetylene **2**: 117 mg, 0.5 mmol; DCM (2 mL)) as a yellow solid, mp 143–145 °C. ^1^H NMR (400 MHz, CDCl_3_): *δ* 1.20–1.31 (m, 2H), 1.67–1.81 (m, 3H), 2.50 (d, *J* = 8.0 Hz, 2H), 2.92–3.01 (m, 2H), 4.02–4.25 (m, 2H), 6.32 (br. s, 1H), 6.56 (s, 1H), 7.04 (d, *J* = 7.4 Hz, 2H), 7.12 (t, *J* = 7.6 Hz, 1H), 7.18–7.28 (m, 7H), 7.36 (t, *J* = 7.6 Hz, 1H), 7.51 (d, *J* = 6.9 Hz, 2H), 7.72 (d, *J* = 7.4 Hz, 2H), 8.69 (br. s, 1H). ^13^C NMR (100 MHz, CDCl_3_): *δ* 31.9, 32.0, 37.6, 42.7, 48.4, 48.5, 68.7, 91.9, 100.8, 117.2, 125.0, 126.5, 128.28, 128.34, 128.5, 128.6, 128.7, 129.2, 132.0, 139.2, 139.4, 141.4, 160.5, 170.1, 188.6. IR (ATR, ZnSe, cm^−1^): *ν* 3417, 3169, 3060, 3025, 2919, 2851, 2191, 1623, 1596, 1571, 1517, 1491, 1451, 1429, 1401, 1384, 1326, 1299, 1223, 1198, 1178, 1156, 1085, 1069, 1051, 1025, 1001. HRMS (ESI-TOF) *m*/*z*: [M + H]^+^ Calcd. for C_31_H_30_N_3_O_2_ 476.2332; Found: 476.2332.

*(Z)-2-(Azepan-1-yl)-5-hydroxy-4-(2-oxo-2-phenylethylidene)-5-phenyl-4,5-dihydro-1H-pyrrole-3-carbonitrile* (**6e**). Compound **6e** was obtained at a 92% yield (174 mg), according to the general procedure (amidine **3e**: 82 mg, 0.5 mmol; acetylene **2**: 117 mg, 0.5 mmol; DCM (2 mL)) as a yellow solid, mp 212–214 °C. ^1^H NMR (400 MHz, CDCl_3_): *δ* 1.50–1.83 (m, 8H), 3.36–3.91 (m, 4H), 6.52 (s, 1H), 6.67 (br. s, 1H), 7.17–7.29 (m, 5H), 7.34 (t, *J* = 7.6 Hz, 1H), 7.49 (d, *J* = 8.3 Hz, 2H), 7.72 (d, *J* = 8.3 Hz, 2H), 8.58 (br. s, 1H). ^13^C NMR (100 MHz, CDCl_3_): *δ* 26.9, 29.4, 50.6, 68.5, 91.9, 100.2, 117.3, 125.0, 128.2, 128.3, 128.4, 128.6, 131.8, 139.5, 141.3, 161.0, 170.6, 188.3. IR (ATR, ZnSe, cm^−1^): *ν* 3430, 3227, 3060, 2928, 2915, 2859, 2192, 1630, 1595, 1573, 1521, 1493, 1470, 1453, 1437, 1420, 1402, 1385, 1368, 1345, 1309, 1221, 1201, 1177, 1157, 1119, 1101, 1083, 1068, 1051, 1027, 1013. HRMS (ESI-TOF) *m*/*z*: [M + H]^+^ Calcd. for C_25_H_26_N_3_O_2_ 400.2020; Found: 400.2015.

*5-(Z)-5-Hydroxy-2-morpholino-4-(2-oxo-2-phenylethylidene)-5-phenyl-4,5-dihydro-1H-pyrrole-3-carbonitrile* (**6f**). Compound **6f** was obtained at an 86% yield (158 mg), according to the general procedure (amidine **3f**: 76 mg, 0.5 mmol; acetylene **2**: 117 mg, 0.5 mmol; DCM (2 mL)) as a yellow solid, mp 207–209 °C. ^1^H NMR (400 MHz, CDCl_3_): *δ* 3.54–3.69 (m, 8H), 6.17 (br. s, 1H), 6.60 (s, 1H), 7.19–7.29 (m, 5H), 7.37 (t, *J* = 7.3 Hz, 1H), 7.52 (d, *J* = 8.2 Hz, 2H), 7.70 (d, *J* = 7.6 Hz, 2H), 8.88 (br. s, 1H). ^13^C NMR (100 MHz, CDCl_3_): *δ* 47.5, 66.1, 68.5, 91.9, 101.9, 117.1, 125.0, 128.3, 128.4, 128.6, 128.9, 132.2, 139.2, 141.2, 161.2, 169.4, 189.0. IR (ATR, ZnSe, cm^−1^): *ν* 3414, 3196, 3058, 2965, 2922, 2893, 2856, 2191, 1621, 1594, 1569, 1518, 1491, 1454, 1429, 1384, 1353, 1332, 1306, 1288, 1267, 1224, 1201, 1172, 1158, 1115, 1085, 1070, 1048, 1033, 1022, 1002. HRMS (ESI-TOF) *m*/*z*: [M + H]^+^ Calcd. for C_23_H_22_N_3_O_3_ 388.1656; Found: 388.1654.

*(Z)-2-(Cyclohexylamino)-5-hydroxy-4-(2-oxo-2-phenylethylidene)-5-phenyl-4,5-dihydro-1H-pyrrole-3-carbonitrile* (**6g**). Compound **6g** was obtained at a 95% yield (190 mg), according to the general procedure (amidine **3l**: 82 mg, 0.5 mmol; acetylene **2**: 118 mg, 0.5 mmol; DCM (2 mL)) as a yellow solid, mp 216–218 °C. ^1^H NMR (400 MHz, CDCl_3_): 1.11–1.344 (m, 5H), 1.59 (br. s, 1H), 1.75 (br. s, 2H), 1.91–1.99 (m, 2H), 3.27–3.42 (m, 1H), 5.60 (d, *J* = 8.2 Hz, 1H), 6.48 (br. s, 1H), 7.30–7.36 (m, 2H), 7.41–7.44 (m, 1H), 7.55 (d, *J* = 8.0 Hz, 2H), 7.78 (d, *J* = 8.0 Hz, 2H), 10.22 (br. s, 1H). ^13^C NMR (100 MHz, CDCl_3_**)**: 24.6, 24.7, 25.0, 33.2, 33.5, 53.4, 68.2, 93.5, 115.9, 125.2, 128.2, 128.4, 128.5, 128.9, 132.0, 139.4, 140.6, 161.1, 183.4. IR (ATR, ZnSe, cm^−1^): *ν* 3298, 3229, 3182, 2185, 1638, 1575, 1484, 1359, 1325, 1298, 1247, 1117, 1044. HRMS (ESI-TOF) *m*/*z*: [M + H]^+^ Calcd. for C_25_H_26_N_3_O_2_ 400.2020; Found: 400.2018.

*(Z)-2-Amino-1-benzyl-5-hydroxy-4-(2-oxo-2-phenylethylidene)-5-phenyl-4,5-dihydro-1H-pyrrole-3-carbonitrile* (**7a**). Compound **7a** was obtained at a 92% yield (94 mg), according to the general procedure (amidine **3g**: 43 mg, 250 µmol; acetylene **2**: 59 mg, 250 µmol; DCM (1 mL)) as a yellow solid, mp 230–232 °C. ^1^H NMR (400 MHz, CDCl_3_): 4.16 (d, *J* = 16.6 Hz, 1H), 4.49 (d, *J* = 16.6 Hz, 1H), 5.11 (br. s, 1H), 6.55 (s, 1H), 7.17 (d, *J* = 6.6 Hz, 2H), 7.29–7.36 (m, 8H), 7.41–7.46 (m, 1H), 7.66 (d, *J* = 7.4 Hz, 2H), 7.81 (d, *J* = 7.4 Hz, 2H), 9.40 (s, 1H). IR (ATR, ZnSe, cm^−1^): *ν* 3445, 3333, 3275, 3181, 3056, 3027, 2197, 1683, 1674, 1595, 1573, 1543, 1473, 1439, 1383, 1357, 1337, 1325, 1314, 1304, 1296, 1258, 1215, 1201, 1172, 1155, 1126, 1109, 1076, 1059, 1026, 1003. HRMS (ESI-TOF) *m*/*z*: [M + H]^+^ Calcd. for C_26_H_22_N_3_O_2_ 408.1707; Found: 408.1706.

*5-(Z)-2-Amino-5-hydroxy-4-(2-oxo-2-phenylethylidene)-5-phenyl-1-(prop-2-yn-1-yl)-4,5-dihydro-1H-pyrrole-3-carbonitrile* (**7b**). Compound **7b** was obtained at a 76% yield (136 mg), according to the general procedure (amidine **3k**: 60 mg, 0.5 mmol; acetylene **2**: 117 mg, 0.5 mmol; DCM (2 mL)) as a yellow solid, mp 187–187 °C. ^1^H NMR (400 MHz, CDCl_3_): *δ* 2.29 (s, 1H), 3.85 (dd, *J* = 18.4, 2.5 Hz, 1H), 4.02 (dd, *J* = 18.4, 2.5 Hz, 1H,), 6.07 (br. s, 2H), 6.52 (s, 1H), 7.30–7.41 (m, 5H), 7.41–7.45 (m, 1H), 7.59 (d, *J* = 7.6 Hz, 2H), 7.79 (d, *J* = 7.6 Hz, 2H), 9.30 (s, 1H). ^13^C NMR (100 MHz, CDCl_3_): *δ* 30.4, 68.1, 74.7, 76.0, 95.5, 100.8, 116.0, 125.8 128.2, 128.3, 129.1, 132.0, 138.3, 139.4, 162.6, 167.6, 188.7. IR (ATR, ZnSe, cm^−1^): *ν* 3432, 3340, 3288, 3263, 3195, 2190, 1663, 1613, 1597, 1576, 1546, 1498, 1477, 1450, 1423, 1385, 1343, 1322, 1304, 1297, 1235, 1211, 1175, 1156, 1106, 1078, 1059, 1033, 1025, 1017, 1003. HRMS (ESI-TOF) *m*/*z*: [M + H]^+^ Calcd. for C_22_H_18_N_3_O_2_ 356.1394; Found: 356.1391.

### 3.2. X-ray Structure Determination

**4e**: Crystal Data for C_14_H_17_N_3_O_3_ (M = 275.30 g/mol): monoclinic, space group P21/c (no. 14), a = 5.5382(12) Å, b = 23.708(6) Å, c = 10.245(2) Å, β = 91.08(2)°, V = 1344.9(5) Å^3^, Z = 4, T = 295(2) K, μ(MoKα) = 0.71073 mm^−1^, Dcalc = 1.360 g/cm^3^, 6120 reflections measured (2.166° ≤ Θ ≤ 29.683°), 3119 unique (Rint = 0.0719, Rsigma = 0.0942), which were used in all calculations. The final R1 was 0.0823 (I > 2σ(I)), and wR2 was 0.2489 (all data).

**6e**: Crystal Data for C_25_H_25_N_3_O_2_ (M = 399.48 g/mol): orthorhombic, space group Pna21 (no. 33), a = 13.549(3) Å, b = 19.212(6) Å, c = 8.2516(15) Å, V = 2147.8(9) Å^3^, Z = 4, T = 295(2) K, μ(MoKα) = 0.71073 mm^−1^, Dcalc = 1.235 g/cm^3^, 10,842 reflections measured (2.120° ≤ Θ ≤ 29.465°), 4991 unique (Rint = 0.0390, Rsigma = 0.0564), which were used in all calculations. The final R1 was 0.0494 (I > 2σ(I)), and wR2 was 0.1232 (all data).

**7a**: Crystal Data for C_26_H_21_N_3_O_2_ (M = 407.46 g/mol): monoclinic, space group P21/c (no. 14), a = 13.216(4) Å, b = 9.084(3) Å, c = 18.091(4) Å, β = 91.62(2)°, V = 2170.9(10) Å^3^, Z = 4, T = 295(2) K, μ(MoKα) = 0.71073 mm^−1^, Dcalc = 1.247 g/cm^3^, 11,891 reflections measured (2.766° ≤ Θ ≤ 29.723°), 5137 unique (Rint = 0.0590, Rsigma = 0.0946), which were used in all calculations. The final R1 was 0.0656 (I > 2σ(I)), and wR2 was 0.1904 (all data).

The experiment was accomplished on the automated X-ray diffractometer «Xcalibur 3» with CCD detector following standard procedures (MoKα-irradiation, graphite monochromator, ω-scans with 1o step at T = 295(2) K). Empirical absorption correction was applied. The structure was solved using the intrinsic phases in the ShelXT program [24] and refined by ShelXL [25] using a full-matrix least-squared method for non-hydrogen atoms. The H-atoms were placed in the calculated positions and were refined in isotropic approximation. The solution and refinement of the structures were accomplished with the Olex program package [26].

CCDC 2254801 (**4e**), 2254802 (**6e**), and 2244496 (**7a**) contains the Appendix A for this paper. These data can be obtained free of charge via http://www.ccdc.cam.ac.uk/conts/retrieving.html (accessed on 17 April 2023) (or from the CCDC, 12 Union Road, Cambridge CB2 1EZ, UK; Fax: +44 1223 336033; E-mail: deposit@ccdc.cam.ac.uk).

## 4. Conclusions

In order to develop an efficient method for the synthesis of aromatic pyrroles, the reaction of DMAD (**1**) and dibenzoylacetylene (**2**) with 3,3-diaminoacrylonitriles **3** was studied. It was shown that the reaction between these compounds proceeds smoothly in dichloromethane with the formation of functionalized nonaromatic pyrroles containing amino, cyano and hydroxy groups, as well as exocyclic C=C and C=O bonds, in high yields.

A revision of the structure of compounds obtained by Cocco and colleagues [18] in the reaction of DMAD (**1**) with 3,3-diaminoacrylonitriles **3a**–**m** was carried out, and based on 2D HMBC NMR and HRMS spectroscopy data, it was concluded that in the studied reaction rather 2-oxo-1*H*-pyrrol-3(2*H*)-ylidenes than 2(1*H*)-pyridones are formed.

## Data Availability

Data are contained within the article and Appendix A.

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
