# Peer review of "Synthesis of Non-Aromatic Pyrroles Based on the Reaction of Carbonyl Derivatives of Acetylene with 3,3-Diaminoacrylonitriles"

_molecules, 2023, doi:10.3390/molecules28083576_

Round 1

Reviewer 1 Report

Bakulev and co-workers reported the reaction of 3,3-diaminoacrylonitriles with DMAD and 1,2-dibenzoylacetylene for the synthesis of non-aromatic pyrroles. There are some important issues to be addressed before the acceptance in Molecules.

1.      Are 3,3-diaminoacrylonitriles 3a-3m all reported? If yes, please provide the reference, or please provide the characterization data including 1H and 13C NMR.

2.      The structures of products 6 and 7 in Scheme 4 and 5 are different in the configuration of double bond, please correct.

3.      The configuration of double bond between products 4, 5 and 6, 7 is different, please give the explanation.

4.      HRMS of 4c and 5b have some errors, please check.

5.      In SI, 4q in Fig. S16 and S17 should be 4g.

6.      Please provide the 13C NMR of product 7a.

7.      1H NMR of products 4d, 5c, 5f, 6b and 6e in SI are different with the corresponding spectra data in manuscript, please correct.

Author Response

1. Are 3,3-diaminoacrylonitriles 3a-3m all reported? If yes, please provide the reference, or please provide the characterization data including 1H and 13C NMR.

We have added the necessary information to Materials and Methods section.

Preparation of 3,3-diaminoacrylonitriles 3. 3,3-Diaminoacrylonitriles 3a–f, h, l were synthesized from ethyl 2-cyanoacetimidate and corresponding amines according to the literature procedures [20-23], compounds 3g,i-k,m are commercially available.

References section is also updated:

Cocco, M.T.; Congiu, C.; Onnis, V.; Maccioni, A. Synthesis of 2-Amino-5-pyrimidinecarbonitrile Derivatives. Synthesis 1991, 529–530. DOI: 10.1055/s-1991-26508

Shafran, Y.M.; Silaichev, P.S.; Bakulev, V.A. β-(Cycloalkylamino)ethanesulfonyl azides as novel water-soluble reagents for the synthesis of diazo compounds and heterocycles. Chem. Heterocycl. Compds. 2019, 12, 1251–1261. DOI: https://doi.org/10.1007/s10593-019-02609-z

Silaichev, P.S.; Beryozkina, T.V.; Novikov, M.S.; Dehaen, W.; Bakulev, V.A. A Base-Controlled Reaction of 2-Cyanoacetamidines (3,3-Diaminoacrylonitriles) with Sulfonyl Azides as a Route to Nonaromatic 4-Methylene-1,2,3-triazole-5-imines. Eur. J. Org. Chem. 2020, 3688–3698. DOI: doi.org/10.1002/ejoc.202000453

Silaichev, P.S.; Beryozkina, T.V.; Ilkin, V.G.; Novikov, M.S.; Dehaen, W.; Bakulev, V.A. Tandem Cycloaddition of Azides to 3,3-Diaminoacrylonitriles (2-Cyanoacetamidines) and Cornforth Type Rearrangement as an Approach to 5-Amino-1,2,3-triazole-4-N-substituted Imidamides. J. Org. Chem. 2023, accepted to publication.

2. The structures of products 6 and 7 in Scheme 4 and 5 are different in the configuration of double bond, please correct.

Structures 6 and 7 were corrected.

3. The configuration of double bond between products 4, 5 and 6, 7 is different, please give the explanation.

Configuration of pairs of isomers around double bond 4, 5 and 6, 7 was confirmed by the data of X-ray analysis. We added new data for compound 4e. The reason for geometrical isomerism could be the formation of different types of hydrogen bonds. Thus in compounds 4 and 5 H-bond is formed from interaction of oxygen of exocyclic C=O bond with H atom at exocyclic C=C bond. While in compounds 6 and 7, H-bond is formed by interaction of oxygen of ketone group with H atom of OH group.

4. HRMS of 4c and 5b have some errors, please check.

Calculated MW for 4c and 5b was corrected.

5. In SI, 4q in Fig. S16 and S17 should be 4g.

Corrected

6. Please provide the 13C NMR of product 7a.

The structure of compound 7a was reliably confirmed by 1H NMR, IR, HRMS and by Х-ray data analysis. We made re-synthesis of this compound three times. But all our attempts to record good carbon spectra failed because this compound turned out to be instable in DMSO and poorly soluble in other solvents. Perhaps you will find our data is good enough to confirm the structure of pyrrole 7a.

7. 1H NMR of products 4d, 5c, 5f, 6b and 6e in SI are different with the corresponding spectra data in manuscript, please correct.

Minor inaccuracies have been corrected both in the text and in SI.

Reviewer 2 Report

In this manuscript, Bakulev and co-works developed the synthesis of substituted pyrrole derivatives through the cycloaddtion reaction of carbonyl derivatives of acetylene with 3,3-diaminoacrylonitriles. The authors proposed the interaction of 3,3-diaminoacrylonitriles with 1,2-dibenzoylacetylene could result in different products, the mechanistic proposal has been provided. The references cited by the authors are appropriate and important. However, there are some mistakes in the manuscript that the authors should correct. Given of these, I would like to accept this manuscript after a minor modification has been made.

There are some mistakes in the manuscript, For example:

1) Scheme 3 and 5: a full arrow should be used to illustrate the nucleophilic attack from species A to alkyne derivatives.

2) How about the reactivity of different electron withdrawing group like nitro of substrate 3? The different substituents on the phenyl group of substrate 2 can affect the reaction dramatically or not?

3) The HRMS of 6d is wrong, please check it.

4) The 13C NMR data of 7a is missing, please check it.

Author Response

1) Scheme 3 and 5: a full arrow should be used to illustrate the nucleophilic attack from species A to alkyne derivatives.

Done

2) How about the reactivity of different electron withdrawing group like nitro of substrate 3? The different substituents on the phenyl group of substrate 2 can affect the reaction dramatically or not?

We have no experimental data on reactivity of electron withdrawing group like nitro group. We only can speculate how introduction of this group can affect the reactivity of the starting compounds. I propose no effect in the case of compounds 2. Contrary reactivity of compounds 3 should be dramatically increased by introduction of nitro group.

3) The HRMS of 6d is wrong, please check it.

We have corrected the data of the spectrum.

4) The 13C NMR data of 7a is missing, please check it.

The structure of compound 7a was reliably confirmed by 1H NMR, IR, HRMS and by Х-ray data analysis. We made re-synthesis of this compound three times. But all our attempts to record good carbon spectra failed because this compound turned out to be instable in DMSO and poorly soluble in other solvents. Perhaps you will find our data is good enough to confirm the structure of pyrrole 7a.

Reviewer 3 Report

This research article by Prof. Vasiliy A. Bakulev and coworkers describes the reaction of 3,3-diaminoacrylonitriles with DMAD or 1,2-dibenzoylacetylene for synthesis of non-aromatic pyrroles which containing exocyclic C=C bond and sp3 hybrid carbon. The formed non-aromatic pyrroles depend on the structure of both acetylene and diaminoacrylonitrile. The substrate scope includes 22 examples, and the author proposed corresponding mechanisms for the reactions. But the manuscript is poorly organized, formats were not uniform and careless ChemDraw drawing in mechanism scheme (even some obvious mistakes). I feel that the manuscript is not suitable for Molecules. Publication in a more specialized journal is recommended after the following revisions (Re-writing the article is strongly recommended):

1.     Lines 41-44: The author said “Several years ago…” but the reference is published in 1998, that’s not several ago (it’s 25 years ago!). The author said, “Then this reaction was used to prepare other types of…”, that’s senseless because the reported method is “form 2-oxoethylidene-4-oxothiazolidin-5-ylidenes”, how to use it prepare “other types” compounds?

2.     Line 41: “thioamides of malonic acid” should be “malonthioamide”.

3.     The author should move Figure 1 to supporting information. There is no need to show starting materials in a separated figure.

4.     Line 75: “Structures of Starting …” should be “Structures of Starting Materials…”

5.     In Scheme 2, the author should symmetrically arrange the products instead of the first line having 4 compounds, the second line 3 compounds and the last line 6 compounds.

6.     In Scheme 2, the author should show more details of the reaction conditions, such as reactants equivalent and concentration.

7.     In scheme 3, 1) the half arrow used between A and 1 is wrong, half arrow is specific used for single electron transfer, here should use the full arrow, 2) C cannot rotational isomerization to D directly, because the C=C bond locked the conformation, the reasonable process should be intermediate B rotational isomerization followed by H-shift. 3) The arrow in intermediate D should be carefully drawn. Same for Scheme 5.

8.     Line 138, “3a-g,k,l should be 3a-g, k, l”, please add space between the numbers.

9.     In “3.1 Synthesis”: All compound’s written in italics due to carelessness.

10.  In the Reference, the author should uniform the formats, for example 1) the reference title some is capitalize every word and some not, 2) references 3, 4 should italicize the volume numbers, 3) reference 6, should remove “x”, 3) reference 16 even forgot put reference title…

11.  The English of the manuscript needs further polish. 

Author Response

1. Lines 41-44: The author said “Several years ago…” but the reference is published in 1998, that’s not several ago (it’s 25 years ago!).

The text was correct.

The author said, “Then this reaction was used to prepare other types of…”, that’s senseless because the reported method is “form 2-oxoethylidene-4-oxothiazolidin-5-ylidenes”, how to use it prepare “other types” compounds?

It was corrected in the text for: This reaction was also used to prepare various derivatives of 4-oxothiazolidin-5-ylidenes [1-3].

2. Line 41: “thioamides of malonic acid” should be “malonthioamide”.

Done

3. The author should move Figure 1 to supporting information. There is no need to show starting materials in a separated figure.

We would like to leave Figure 1 in the main text, for authors convenience.

4. Line 75: “Structures of Starting …” should be “Structures of Starting Materials…”

Done

5. In Scheme 2, the author should symmetrically arrange the products instead of the first line having 4 compounds, the second line 3 compounds and the last line 6 compounds.

Done

6. In Scheme 2, the author should show more details of the reaction conditions, such as reactants equivalent and concentration.

Done

7. In scheme 3, 1) the half arrow used between A and 1 is wrong, half arrow is specific used for single electron transfer, here should use the full arrow, 2) C cannot rotational isomerization to D directly, because the C=C bond locked the conformation, the reasonable process should be intermediate B rotational isomerization followed by H-shift. 3) The arrow in intermediate D should be carefully drawn. Same for Scheme 5.

1) The half arrow was changed to full arrow.

2). We redraw the Schemes 3 and 5 according to your offer.

3) Done

8. Line 138, “3a-g,k,l should be 3a-g, k, l”, please add space between the numbers.

Done

9. In “3.1 Synthesis”: All compound’s written in italics due to carelessness.

Corrected

10. In the Reference, the author should uniform the formats, for example 1) the reference title some is capitalize every word and some not, 2) references 3, 4 should italicize the volume numbers, 3) reference 6, should remove “x”, 3) reference 16 even forgot put reference title…

All comments to the list of references are taken into account.

11. The English of the manuscript needs further polish.

We asked a person who knows English perfect to polish the language.

Round 2

Reviewer 3 Report

Prof. Vasiliy A. Bakulev and coworkers reported the reaction of 3,3-diaminoacrylonitriles with DMAD or 1,2-dibenzoylacetylene for synthesis of non-aromatic pyrroles which containing exocyclic C=C bond and sp3 hybrid carbon. The formed structure of non-aromatic pyrroles depends on the structure of both acetylene and diaminoacrylonitrile. The scope of the method is highlighted in 22 examples, and the author proposed corresponding mechanisms for the reactions. Authors also given full modifications in the revised manuscript submitted to the Molecules. I feel that the revised manuscript deserved to be published in Molecules.